# Pesticide Exposure in Fruit-Growers: Comparing Levels and Determinants Assessed under Usual Conditions of Work (CANEPA Study) with Those Predicted by Registration Process (Agricultural Operator Exposure Model)

**DOI:** 10.3390/ijerph19084611

**Published:** 2022-04-11

**Authors:** Morgane Bresson, Mathilde Bureau, Jérémie Le Goff, Yannick Lecluse, Elsa Robelot, Justine Delamare, Isabelle Baldi, Pierre Lebailly

**Affiliations:** 1ANTICIPE, INSERM U1086, Centre François Baclesse, University of Caen Normandie, 14000 Caen, France; j.le-goff@baclesse.unicancer.fr (J.L.G.); y.lecluse@baclesse.unicancer.fr (Y.L.); 21300932@etu.unicaen.fr (J.D.); p.lebailly@baclesse.unicancer.fr (P.L.); 2EPICENE, INSERM U1219, Bordeaux Population Health Center, University of Bordeaux, 33076 Bordeaux, France; mathilde.bureau@u-bordeaux.fr (M.B.); elsa.robelot@u-bordeaux.fr (E.R.); isabelle.baldi@u-bordeaux.fr (I.B.)

**Keywords:** pesticides, occupational exposure, exposure model, fruit-growing, pesticide registration

## Abstract

Knowledge of pesticide exposure levels in farmers is necessary for epidemiological studies and regulatory purposes. In the European pesticide registration process, operators’ exposure is predicted using the Agricultural Operator Exposure Model (AOEM), created in 2014 by the European Food Safety Authority based on studies conducted by the pesticide industry. We compared operators’ exposures during treatment days in the apple-growing industry under non-controlled working conditions and AOEM-predicted values. The dermal exposure of thirty French apple-growers from the CANEPA study when applying two fungicides was measured using body patches and cotton gloves. For each observation, the corresponding exposure was calculated by means of the AOEM, using data recorded about the operator, spraying equipment and personal protective equipment (PPE) used. A significant linear correlation was observed between calculated and measured daily exposures. The model overestimated the daily exposure approximately 4-fold and the exposure during application 10-fold. However, exposure was underestimated during mixing/loading for 70% of the observations when the operator wore PPE. The AOEM did not overestimate exposures in all circumstances, especially during mixing/loading, when operators handle concentrated products. The protection provided by PPE appeared to be overestimated. This could be due to the optimal working conditions under which the “industrial” studies are conducted, which may not be representative of real working conditions of operators in fruit-growing.

## 1. Introduction

Pesticide use for crop protection in agriculture developed in the 1930s and increased worldwide in the second part of the 20th century (2.3 billion tons in 1990 vs. 4.1 billion tons in 2018) (FAOSTAT). Europe is a major agricultural producer and pesticide consumer with 500,000 tons sold in 2018, including 85,000 tons for France alone (FAOSTAT). There are currently more than 1000 active substances on the market used in the control of pests, each with different chemical characteristics, targets and modes of action. They can be classified into three main categories: herbicides, insecticides and fungicides. Fungicides were the best-selling pesticide categories in 2018, representing 40% of total pesticides sold in Europe (FAOSTAT) [1]. Numerous epidemiological studies have also shown that the use of pesticides overall, as well as the use of some of these substances, were associated with the occurrence of chronic diseases such as certain cancers [2,3], neurodegenerative diseases [4] and respiratory diseases [5]. Less than one hundred pesticides have been evaluated by the International Agency for Research on Cancer (IARC), and these have been classified into five categories (1, 2A, 2B, 3 and 4) according to the level of evidence of their potential carcinogenicity. Lindane, pentachlorophenol and arsenates were classified as carcinogens for human and less than 10 were classified as probably being carcinogenic, such as DDT and glyphosate. In 2010 in France, more than one million farmers and farm-workers were exposed to pesticides, to which hundreds of thousands of seasonal workers and trainees must be added [6].

Since 1993, the approval of pesticides has been regulated by the European Union and includes toxicological and environmental criteria. Toxicity assessments involve the consideration exposure prediction models that were initiated in the 1980s and 1990s based on national initiatives (UK-POEM in the United Kingdom, BBA in Germany). In 1993, Van Hemmen demonstrated their limitations, namely, that they were based on unpublished and heterogeneous exposure studies with differences in protocols and statistics, and only contained data for some exposure scenarios [7]. He recommended exposure models built on a European scale and based on new studies taking into account a larger range of farm sizes and farming methods [7]. In 2001, the European Commission asked a group of experts to develop an operator exposure database called “EUROPOEM” [8]. In 2010, UK-POEM and BBA were still being used in Europe, even though they were considered to be no longer fully representative of agricultural practices. The European Food Safety Authority (EFSA) found that the risk assessment methodology was no longer adequate, data underlying exposure estimates were scarce for some scenarios, and there were possible inconsistencies between models used by regulatory authorities in different EU member states [9]. In 2014, the EFSA edited a publication entitled *Guidance on the Assessment of Exposure for Operators*, *Workers*, *Residents and Bystanders in Risk Assessment for Plant Protection Products*, in which the Agricultural Operator Exposure Model (AOEM) was described [10]. This was the first European operator exposure model designed for use in pesticide regulation for operators, and it was based on 34 studies conducted between 1994 and 2009 and selected by an expert panel of regulators and industry representatives [10,11]. These studies were provided by the European Crop Protection Association (ECPA) and its member companies, and carried out under controlled conditions in 10 countries (France, Germany, Spain, UK, Portugal, the Netherlands, Italy, Greece, Switzerland and Belgium) [11]. Unfortunately, information available from these studies was limited to some descriptive data [11]. In contrast, operator exposure data produced by academic research teams in different types of crops and with different pesticides are available in various European countries and agricultural settings, such as fruit-growing in the Netherlands [12], open-field farming in France and in Italy [13,14], vineyards in France and Greece [15,16] and tomato-growing in Italy [17]. However, they have not been incorporated into the AOEM. Since 2009, the regulation (EU) 1107/2009 defines the process for pesticide approval, including different components: physico-chemical analytical methods, efficacy, toxicity, residues, fate and behavior, ecotoxicology and crop specificity. The AOEM is one of the tools used in the toxicity section. As the European Commission points out, improvements are still needed for better pest management, safer handling, safer food consumption (residue reduction) and environmental protection, with measures such as the implementation of integrated pest management (IPM), the promotion of alternative approaches or techniques and operator training [18].

The PESTEXPO project was initiated in the 2000s in France to study farmers’ exposure to pesticides under usual conditions of work. The main targeted crops were vine crops [15], open-field crops (wheat, corn) [13], market gardening, weeding in non-agricultural areas and recently fruit-growing [19,20]. In the present study, we compared exposure of operators during pesticide application days, within the framework of the CANEPA (Cancer and Exposure to Pesticides in Agriculture) project, with the corresponding exposure values calculated using the AOEM. We sought to determine whether the AOEM realistically estimated the daily exposure of agricultural operators involved in apple-growing and was correlated with it.

## 2. Materials and Methods

### 2.1. Study Population

The CANEPA study was conducted in four widespread areas in France (Normandy, Rhône-Alpes, Poitou-Limousin, Garonne Valley), over two agricultural seasons (in 2016 and/or 2017), on 24 fruit-growing farms. The study has been described in detail in a previous publication [20]. In brief, overall 107 workers were observed on 156 working days: 30 treatment days, corresponding to 52 mixing/loading, 52 application and 12 cleaning phases observed, and 126 days of re-entry or harvesting tasks in previously treated fields. In the present study, only the 30 operators involved in pesticide applications were considered. They performed treatment tasks, including mixing/loading, application and equipment cleaning. Operators were mainly males (97%) and were farm owners or permanent employees. This implies that the participants in our study had some experience with treatment tasks, had all received agricultural training and obtained the Certiphyto (the mandatory certification for pesticide use, following training in safe pesticide use and reduction) and were likely aware of the risks associated with pesticides. An observation day consisted of observing one operator involved in the mixing/loading phase(s), the application phase(s) and in the cleaning of equipment during their whole day of treatment.

### 2.2. Observations of Farm Operators and Sample Collection

Farm operators were observed and samples collected in order to estimate dermal and inhalation exposures and to determine factors associated with exposure. We focused here on dermal exposure since inhalation exposure is relatively limited compared to dermal exposure [20,21,22,23].

Dermal exposure was measured using the skin pad method because it was more suitable for the work environment studied. It involved no change in the usual conditions of work as most of the workers did not usually wear a coverall especially during application, no interference with usual working clothes and no thermal discomfort or limitation of movement [24]. This method was approved by the Organization for Economic Co-operation and Development (OECD) [25] and has been widely used in fruit-growing exposure studies [12,22,26,27,28,29,30]. Eleven 10 × 10 cm patches were placed on the head (1 patch on a cap) and on different parts of the body directly on the skin: the arms (2 patches), forearms (2 patches), chest (1 patch), back (1 patch), thighs (2 patches) and lower legs (2 patches) (Appendix A, Figure A1). Cotton gloves were also worn under the protective gloves if used to measure hand exposure. Patches and gloves were changed during each mixing/loading, application and cleaning process to obtain the exposure values for each task separately. On three application days, patches were not replaced between mixing/loading and application, so these days were taken into account only for daily exposure analysis. A total of about 1100 patches and 200 gloves were collected during application days, corresponding to 1300 exposure measurements.

The two active substances studied were two fungicides very frequently used in fruit-growing: captan and dithianon. They were both measured in these samples, along with tetrahydrophthalimide (THPI), the major captan degradation product [20].

All information about operators (worker status, experience, level of education, smoking), farms (total area, type(s) of crop(s)), weather conditions on observation days (temperature, wind speed), type of personal protective equipment (PPE) used, equipment characteristics (type of sprayer, age of tractor and sprayer, presence of a cab) and tasks carried out (technical problems, etc.) were collected by field monitors face to face. Observation notebooks, videos and photos were used to record these auxiliary data. All data have been described previously elsewhere [20].

### 2.3. Use of the Agricultural Operator Exposure Model (AOEM)

The AOEM is based on 34 studies, constituting a database of 2900 exposure measurement values from 280 mixing/loading and 344 application phases, including all types of crops. About a quarter of the studies concerned a “high crop tractor/vehicle-mounted or trailed application”, corresponding mainly to fruit-growing and vine-growing. Six hundred and sixty-nine dermal measurements, obtained via whole-body dosimetry, were included in the model for this scenario: 265 for mixing/loading and 404 for application. These 669 dermal measurements corresponded to situations with unprotected and protected hands, body and head [11].

During the development of the AOEM, more than 50 parameters describing the conditions of application were initially studied (not all of them were described in the original article) [11]. Thirteen key factors, considered to be determinants of exposure, were then selected, including seven factors for mixing/loading and eight for application. The formulation type (wettable powder, wettable granules and liquids) and the total amount of active substance used were considered during each phase. The concentration in the product of the active substance used, the number of containers handled, the number of mixing/loading tasks, the equipment used (induction hopper) and the mixing/loading time were determinants for mixing/loading and the concentration of the active substance in the spray mixture, the equipment (cabin/no cabin), the size of the treated area, droplet size, cleaning (performed or not) and the duration of the phase were those for application. Finally, the only considered parameters that were correlated with exposure appeared to be the quantity of the active substance handled, the active substance formulation for mixing/loading (wettable powder, wettable granules, liquids) and the presence of a cab on the tractor for application. It was not explicitly stated whether the other previously selected parameters were used for exposure calculations [11].

The AOEM is available as an Excel program composed of several spreadsheets (http://www.fao.org/pesticide-registration-toolkit/registration-tools/assessment-methods/method-detail/fr/c/1187029/, accessed on 15 July 2021). The data required to calculate operator exposure are the active substance used (name of the molecule, brand name, formulation), the type of crop treated, the maximum rate of active ingredient applied (kg per hectare), the application scenario (indoor or outdoor application), the spraying method (upward or downward), the spraying equipment (sprayer towed by vehicle, manual hand-held sprayer or knapsack sprayer) and personal protective equipment (PPE) if worn during mixing/loading and application (gloves, long working clothes, hood and visor or only a hood or mask). During application, the presence of a cab on the tractor must also be specified. It is possible to mention other parameters such as the percentage of oral, inhaled or dermal absorption of the product and dilution, and the vapor pressure of the active substance. These data are then used to calculate the internal dose of pesticides absorbed, which is outside the scope of our analysis.

We used all the information collected in the CANEPA field study for each operator monitored during an observation day to calculate their theoretical exposure, assessed using the AOEM. Apple-growing corresponded to “pome fruit” in the model. The two fungicides used on apple trees by the applicators were formulated as water-dispersible granules (WGs) for captan and dithianon and also as liquid in soluble concentrate (SC) for dithianon. The rate of active substance applied varied for each observation day and depended on the active substance used. Pesticide application on apple trees ocurred outdoors, the spray was upward and operators used a vehicle-mounted sprayer. Operators were protected with various forms of PPE: we considered coveralls as long working clothes associated with a hood, and the wearing of protective gloves as efficient.

Adjusting these parameters automatically resulted in the estimation of other parameters. The *Operator Outdoor Spray AOEM* worksheet summarizes the application scenario. A default value of the daily treated area (10 ha for pome fruit) was used to calculate the quantity of active substance used during the day.

All this information was used to calculate long-term (75th centile) and acute (95th centile) exposure of hands, body and head (protected or not) and inhalation during mixing/loading and application. The sum of hands, body (chest and back + arms + legs), head and inhalation exposure values during mixing/loading and application gave an exposure value for a treatment day lasting eight hours.

Here is an example of the calculation equation used in the AOEM to calculate exposure (e.g., for long-term exposure of non-protected hands during mixing/loading) as it appears in the Excel cells:«IF *(ISTEXT(INDEX(ay_OpExAOEM;MATCH(sys_KeyOperator&”Hands”;key_AOEM;0);10)); 10^(INDEX(ay_OpExAOEM;MATCH(sys_KeyOperator&”Hands”;key_AOEM;0);7) + (LOG(‘[EFSA_modele_2014.xlsx]Operator Outdoor Spray AOEM’!i_AmoutAS)* ∗ *INDEX(ay_OpExAOEM;MATCH(sys_KeyOperator&”Hands”;key_AOEM;0);8)) + INDEX(ay_OpExAOEM;MATCH(sys_KeyOperator&”Hands”;key_AOEM;0);9)); MAX(INDEX(ay_OpExAOEM;MATCH(sys_KeyOperator&”Hands”;key_AOEM;0);10);(INDEX(ay_OpExAOEM;MATCH(sys_KeyOperator&”Hands”;key_AOEM;0);10)/d_PctExtrapolation)* ∗ *’[EFSA_modele_2014.xlsx] Operator Outdoor Spray AOEM’!i_AmoutAS))*»(1)

This equation can be simplified into:Exposure = «10 «intercept» factor + [log (Amount of active substance applied per day) × «amount applied» factor] + «scenario» factor»(2)

Exposure during mixing/loading and application, and of the different parts of the body, were calculated using this equation. Exposure was calculated using three factors, the values of which depended on the application scenario, the formulation of the active substance, the part of the body considered and the PPE worn: intercept factor (ranging from −0.98 to 3.12 for mixing/loading, and from 0.23 to 6.08 for application), the amount applied factor (ranging from 0.30 to 1.00 for mixing/loading, and from 0.16 to 1 for application) and the scenario factor (ranging from −1 to 1.83 for mixing/loading and from −1.64 to 1.89 for application). The values of these factors are detailed in another worksheet named “Operator_exposure_values_AOEM”. The values of these factors for the “pome fruit” scenario are presented in Appendix B (Table A1). The calculations used to obtain these values are not explained in the “EFSA guidance”.

For each treatment day, operator exposure was measured in the CANEPA study and calculated using the AOEM.

### 2.4. Statistical Analysis

The variables used to calculate AOEM exposure have been described. We described these variables using the median because they were not normally distributed, according to the Shapiro–Wilk test.

The relationship between measured exposure and calculated exposure was plotted and studied via linear regression (*t*-test). Each point of the scatterplot corresponded to one observation day. Each measured value had a calculated AOEM value.

Values were compared with the Wilcoxon test for the overall day (sum of application and mixing/loading exposure values), for mixing/loading and application separately, and for the different parts of the body. On some working days, the operator carried out several cycles of mixing/loading and application. In these situations, contamination values for each phase were summed to obtain the daily mixing/loading and application exposure values, respectively.

All data analyses were performed using R (Version 3.6.1, R Core Team and the R Foundation for Statistical Computing, Vienna, Austria), R Studio (Version 1.2.1335, RStudio, Inc., Boston, MA, USA) and Microsoft Excel 2019 (Version 16.0.10384.20023, Microsoft Corporation, Redmond, WA, USA) software.

## 3. Results

### 3.1. Data Used for Calculation

The amount of captan applied per hectare (ha) (median = 1.52 kg/ha) and per day (median = 10.32 kg/day) was greater than the amount of dithianon WG and SC applied per hectare (median = 0.35 and 0.31 kg/ha, respectively) and per day (median = 1.4 kg/day for both). The median daily treated area was 5 ha (ranging from 1 ha to 18.3 ha). The AOEM calculated the amount of active substance used on the treatment day from the rate of active substance applied per hectare (kg/ha) multiplied by the treated area, fixed at 10 ha/day for the “pome fruit” scenario. The calculated amount of active substance used in a day was greater for captan (median = 15.2 kg/day) than for dithianon WG or SC (3.5 and 3.1 kg/day, respectively) and higher with the model than by observation for 90% of observations (Table 1).

In terms of equipment used, sprayers were pulled (mounted or trailed) by a tractor and only one operator used a self-propelled vehicle. Two tractors did not have a closed cab. The presence of a cab on the tractor reduces the exposure during application, as calculated via the AOEM.

Various forms of PPE were used during mixing/loading and application by operators, with gloves being the most common. Gloves, protective masks or other types of respiratory and head protection were more frequently used during mixing/loading (77% for gloves and 77% for respiratory protection) than during application (13% for gloves and 10% for respiratory protection). Body PPE was the same during mixing/loading and application: long cotton working clothes were worn in 67% of the observations, and chemical protective coveralls (Tyvec^®^, DuPont, Wilmington, NC, USA) over working clothes in only 20%. Other PPE was used, such as aprons.

### 3.2. Example of Observation 1

Appendix C (Figure A2 and Figure A3) shows an example of exposure results given by the AOEM for the “outdoor, upward spraying, vehicle-mounted sprayer” scenario and captan WG spraying. The active substance application rate was 1.44 kg/ha and the assumed area treated in this scenario was 10 ha, so the amount of active substance applied was 14.4 kg/day. The AOEM gave the long-term (75th centile) exposure values of the different body parts and inhalation for mixing/loading and for application, according to the PPE selected. In this example, the operator wore gloves, mask and long working clothes during mixing/loading; he only wore long working clothes during the application stage. Predicted dermal exposure during mixing/loading (0.367 mg/day) was around 40 times lower than exposure during application (15.26 mg/day).

### 3.3. Comparison of Measured and Predicted Values

#### 3.3.1. Measured Exposure

The median daily exposure was 4.25 mg/day (Figure 1). Exposure during mixing/loading (median = 1.36 mg/day) was comparable to exposure during application (median = 1.27 mg/day) (Figure 2). Hands (median = 1.87 mg/day) accounted for about 40% of the overall daily exposure (Figure 1), with equivalent median values for mixing/loading and application (median = 0.32 mg/day) corresponding to about 25% of the exposure for each phase (Figure 2). However, hand exposure values were very variable from one operator to another (range from 0.001 mg/day to 13.39 mg/day) (Figure 1). The body (median = 0.90 mg/day) contributed slightly less than hands to the daily exposure (Figure 1). Head exposure was measured for mixing/loading and for application for 18 observations (60% of the observations) and was lower (median = 0.22 mg/day) (Figure 1), representing 3%–8% of the total measured exposure. Thus, when not measured (for 12 observations for mixing/loading and for nine observations for application), we considered the head exposure level to be zero. A sensitivity analysis revealed that head exposure had a very low impact on the daily measured exposure (Appendix D, Figure A4 and Figure A5). Furthermore, we did not find any difference in daily exposure levels between farm owners and permanent employees.

#### 3.3.2. Calculated Exposure

Using the AOEM, the median daily exposure was determined to be 15.93 mg/day and was four-fold significantly higher than that measured in the CANEPA study, but with an overlapping distribution between measured and calculated values for all situations (Figure 1). Exposure during application (median = 12.75 mg/day) (10-fold significantly higher than that measured) was higher than during mixing/loading (median = 0.37 mg/day) (not significantly lower than that measured) (Figure 2). The AOEM attributed more than 80% of overall daily exposure to hands (median = 12.97 mg/day), and specifically to exposure of the hands during application (median = 12.75 mg/day) (Figure 1 and Figure 2). The body (median = 1.20 mg/day) and the head (median = 0.26 mg/day) contributed less to the total daily exposure (Figure 1). The same ranking of body parts was found for application separately, although exposure of the hands, body and head were of the same order of magnitude for mixing/loading (medians between 0.08 and 0.20 mg/day) (Figure 2).

### 3.4. Relationship between Measured and Calculated Exposures

There was a positive linear correlation between calculated and measured exposures (r = 0.76, *p*-value < 0.0001) (Figure 3). Overall, the AOEM overestimated daily exposure. However, in four observations (9, 11, 12 and 28), the measured value was higher than the calculated one, whereas they were equivalent in another observation (observation 3).

Calculated and measured exposures during mixing/loading for hands, body or total (body + hands + head) were linearly correlated (r = 0.69, *p*-value < 0.0001) (Figure 4). When considering exposure of the total body (hands + body + head) during mixing/loading separately, the calculated daily values were equal or lower than the measured values in all observations but nine (observations 7, 15, 16, 18, 19, 23, 25, 27 and 29) (Figure 4a). This underestimation was observed for body exposure in all observations but seven (observations 4, 7, 14, 18, 19, 25 and 27). These observations corresponded to operators who did not wear long working clothes (Figure 4c). Results were more varied for hand exposure. Hand exposure was overestimated in 11 operators and underestimated in others, irrespective of the wearing of gloves. The calculated exposure of operators who did not wear gloves was always higher than that of operators wearing them. Measured exposure values of non-gloved operators’ hands were generally among the highest, although one operator (observation 7) had an intermediate exposure value and some operators wearing gloves had high values (observations 1 and 11) (Figure 4b).

Exposure during application was overestimated by the AOEM for all observations but four (observations 9, 12, 28 and 29). No linear correlation was observed (r = 0.22, *p*-value = 0.28) (Figure 5a). Body exposure tended to be overestimated by the AOEM, except for one observation (observation 12), in which exposure was greatly underestimated, and no correlation was observed (r = 0.06, *p*-value = 0.76). The seven observations with the highest calculated exposure (observations 4, 7, 14, 18, 19, 25 and 27) were in operators who did not wear long working clothes (Figure 5c). Concerning hand exposure, we found a lower linear correlation than during mixing/loading between measured and calculated exposure (r = 0.40, *p*-value = 0.04). Hand exposure was overestimated by the AOEM, except in observations 26 and 28. Exposure values were very heterogeneous in terms of measured exposure, whereas there were two levels of values for calculated exposure: between 0.1 and 1 mg and about 10 mg, corresponding to operators wearing gloves or not, respectively (Figure 5b).

The daily treated area was fixed at 10 ha/day according to the AOEM in the “pome fruit” scenario, whereas it was often less than 10 ha in CANEPA study. We manually modified the rates of active substance applied (kg/ha) in the AOEM “data entry” worksheet to obtain the actual amount of active substance handled on the observation day, considering the actual treated area. Thus, calculated exposure decreased after this modification, except in three operators who applied fungicides over more than 10 ha (observations 2, 8 and 10). Daily exposure and application exposure were then less overestimated (83% of the observations initially overestimated and 70% of the observations secondarily overestimated; 85% of the observations initially overestimated and 81% of the observations secondarily overestimated, respectively) and mixing/loading exposure was more underestimated after these modifications (67% of the observations initially underestimated and 70% of the observations secondarily underestimated) (Appendix E, Figure A6).

## 4. Discussion

### 4.1. Main Results

In a context in which few data are available on pesticide exposure in fruit-growers, we produced independent academic results and compared the levels obtained in field observations to those predicted using the regulatory AOEM model. Overall, exposures measured in our study were lower than those calculated by means of the AOEM, although the distribution of the calculated and measured exposures mostly overlapped.

However, some inconsistencies were observed when comparing exposure levels according to tasks (mixing/loading, application) and considering PPE; thus, the model does not overestimate exposures in all circumstances. Specifically, the model appears to overestimate the protection provided by gloves and coveralls and underestimates the contaminations that occur during mixing/loading.

### 4.2. Limitations and Strengths

Since our participants were volunteers, our population was not fully representative of French fruit-growers. Moreover, the average utilized agricultural area of the farms in which our observations were made, i.e., approximately 70 ha in total and 36 ha dedicated to fruit-growing, was higher than that of other French farms (for fruit specialization, average utilized agricultural Area = 16 ha) [31]. We therefore assume that they had larger fields and potentially used newer sprayers and tractors. In addition, the participants were likely more aware of the safer use of pesticides because they received agricultural training and were all Certiphyto holders, compared to the seasonal workers, family helpers, or trainees who mainly perform re-entry tasks (pruning, thinning, harvest). However, they came from different regions of France and fungicide application was carried out in a range of different conditions governed by the weather, types of farms, equipment, and socio-economic backgrounds. Furthermore, we made many observations and took even more samples and exposure measurements.

Although the participants were advised to proceed as usual, they may have changed their practices due to the presence of observers, a situation that would in fact tend to increase their attention and decrease measured exposure. The actual farming practices we observed were sometimes far from the so-called but not clearly defined “good farming practices”, which constitute the basis for the AOEM [10,11]. However, our goal was to capture real-life situations and not to interfere with routine practices.

For body exposure measurement, we used the patch method, one of the OECD-approved techniques (patches and coveralls). However, over the years, whole-body dosimetry seems to have gained the preference of pesticide companies and regulatory agencies. We chose the patch method for several reasons. The main reason was to respect routine working conditions, including the wearing (or not) of PPE. Moreover, observations took place on days in the warm season where the outside temperature reached 34.5 degrees in the shade. Wearing coveralls would have caused discomfort in terms of movement and temperature due to the extra layer of clothing. The patch technique also requires fewer solvents and is less time-consuming, less expensive and more sustainable than whole-body dosimetry [24,32]. In fact, its use in AOEM studies may create a bias in the comparison between measured and calculated exposure. However, the risk of overestimation (splashes on the patch) or underestimation (droplets missing the patch) with the patch technique [24,32] was mitigated by careful observation throughout the sampling period. Furthermore, because of the risk of overestimating exposure, the patch method could also be considered appropriate for “worst-case” exposure estimates [32]. Kasiotis et al. compared the two methods when performing different tasks under standardized conditions in a test chamber. They observed no significant difference among values when exposure measured on the patches was extrapolated to body surface area; therefore, they could not establish a “gold standard” for dermal exposure [32].

### 4.3. External Validity

The AOEM’s calculations are based on the following parameters for the scenario corresponding to the application of pesticide on apple trees: the amount of active substance applied, the formulation, the wearing of PPE and the presence of a cab on the tractor.

The amount of active substance applied per day was a crucial parameter in calculating exposure. It was nearly three-fold higher in the AOEM in comparison to field observations, in relation with a 10 ha reference value for an area treated daily in the AOEM [10,11]. Indeed, fruit-growers in our study treated smaller surfaces (median = 5 ha, 75th percentile = 6.9 ha). Thus, the overestimation of daily exposure by the AOEM was partly due to the larger surface area considered. Wong et al. also found that the daily median treated area was less than 10 hectares for orchards in the UK, Lithuania and Greece, but this value was often exceeded in UK orchards, where newer and more efficient sprayers were used [33]. It was also higher in a study in French vineyards, where the median value of the daily treated area (10.2 ha/day) was equal to the 75th percentile value set by the AOEM (10 ha/day) [15]. This 10-hectare default value thus appears questionable according to the countries and the fruit under consideration.

In our study, all operators but six used the WG formulation, so we cannot compare exposures according to the formulation. According to the AOEM, the granular formulation was associated with the lowest exposures, followed by the liquid and powder formulations. Several scientific publications also found that the formulation of the active substance impacted exposure and that WG forms were the least exposing [21,22,28,33,34,35]. Thus, in the CANEPA study, we may not have observed the worst-case scenario regarding the type of formulation and we do not know whether the underestimation by the AOEM that we observed in some circumstances would have been smaller or larger if liquids or powder had been used. However, the AOEM is not exhaustive on existing formulations, especially the most recent ones, which could have an impact on operator exposure and on PPE protection factors [36].

In the CANEPA study, individuals not wearing PPE were among the most contaminated during mixing/loading, whereas this was less obvious during spraying, a task during which the wearing of gloves was rarely observed. However, PPE heavily influenced the values predicted by the model and resulted in a strong divergence between predicted and observed values. Underestimation by the model was particularly observed among operators wearing gloves and long clothes during mixing/loading. However, the PPE proposed in the AOEM poses some problems, as it does not always correspond to the real field situations. In our field study, trousers were sometimes worn with a top covering the arms, and some operators wore coveralls over their working clothes. Exposure was therefore calculated in the same way for operators wearing only long working clothes that were not specific for chemical protection and for those wearing coveralls over them. In the study by Großkopf et al., working clothes reduced body exposure by 85–98% depending on the scenario, and protective gloves reduced it by 89–99% compared to bare hands during mixing/loading and application [11]. In total, wearing gloves or long working clothes reduced mixing/loading exposure by half, and the combination of both reduced exposure by about 90% for application and by over 95% for mixing/loading. Wearing a mask or hood had little impact on exposure. These protective factors were the same in previous deterministic models (BBA and UK-POEM models) [34]. Hines et al. found that hand exposure during mixing/loading was reduced by 77% by wearing chemical-resistant gloves [28]. These protective factors appeared to have less of an effect in the CANEPA population and the AOEM overestimated the protection levels of PPE compared to the actual protection levels of CANEPA—by three-fold for gloves and about 30-fold for long working clothes. In pesticide application situations, many parameters play a role in the acceptance of PPE and modify their protective efficacy. Weather conditions such as heat create discomfort, hamper thermoregulation [36,37] and increase transpiration, which in turn potentiate the penetration of substances through PPE and the skin [30,34,36,37]. Another factor is a lack of knowledge or training in the use of PPE (gloves not suitable for chemicals, reuse of single-use PPE, reusable PPE contaminated, removal of gloves without taking care not to touch the outside) [34,36,38], and the feeling of security provided by PPE leading to risky behavior [34]. In fact, actual field conditions may hamper the proper use of PPE [36].

Regarding spraying equipment, all but two tractors in our observation were equipped with a cab, so exposure levels could not be compared. However, the proportion of tractors with enclosed cabs in the CANEPA study (93%) was higher than in the studies included in the model for fruit-growing (56%), which could contribute to the model’s overestimation of exposure.

Several other parameters may vary during a treatment day and across farming populations in Europe, such as weather, worker status, the type of fruits grown, the size of orchards and the equipment used [33], and this can affect exposure in fruit-growers [30]. Furthermore, exposure may be operator-dependent, with factors such as age and educational level playing a role [28,30,34]. Exposure also increases with the number of tasks performed daily [34,38] and the number of treatment days in a season. For example, spraying pesticides repeatedly can lead to cross-contamination [12,35,39,40]. The cleaning of equipment at the end of the application day was also a parameter increasing exposure to a greater or lesser extent (med = 0.75 mg/day, min = 0.10 mg/day, max = 44.24 mg/day in the CANEPA study). However, cleaning was not always performed. We excluded it from the analyses because it was not used as a modeling factor [10,11], and we do not know how it was included in the model calculations. In addition, an operator may not receive the same amount of pesticide depending on the equipment used (mounted or trailed sprayer), its age and whether they carry out repairs on the sprayer on the treatment day [15,28,30]. In the agricultural health study (AHS) in the USA, an algorithm based on a literature review included repair tasks as a major determinant of the exposure score (EXPOSURE SCORE = (MIX + APPLY + REPAIR) × PPE) [29]. Equipment cleaning was the final procedure on the application day in 12 observations in the CANEPA study. Großkopf explained that cleaning was not considered as a modeling factor since its impact on exposure had not been demonstrated in studies included in the AOEM [11]. However, owing to contact with the contaminated surfaces of the spraying equipment [39], this final step has proven to create significant exposure in operators [15], increasing urinary levels of metabolites [27,38]. Overall, it was even found to be the phase with the most exposure in the CANEPA study [20].

The AOEM is intended to provide conservative exposure estimations, which means increasing protection for operators. Calculated exposure must be higher than actual exposure. To estimate operator exposure, the AOEM used the 75th percentile value, which is assumed to provide a rarely-exceeded estimate of daily exposure [9]. However, this means that on a quarter of spraying days, actual exposure should be higher than estimated exposure. The AOEM also overestimates the daily treated area (10 ha versus a median of 5 ha in our observations) to obtain a daily exposure value that is slightly higher than if the actual treated area is taken into account. However, this did not avoid underestimating mixing/loading exposure, a phase that lasted only 15 min on average (vs. 77 min for spraying) but which involves even more exposure as the product handled is in a concentrated form [30,33,39]. In addition, the daily treated area was greater than 10 ha in three observations. The equipment and machines used on smaller farms can reasonably be expected to be older, less efficient and less safe, allowing a smaller area to be treated in a day and involving a high level of exposure. In some cases of PPE use, its protective effect was overestimated, leading to an underestimation of operators’ exposure during mixing/loading in orchards. Wearing PPE under real uncontrolled conditions was not always associated with low exposure because protection was not always as effective as expected.

## 5. Conclusions

Determining the exposure of agricultural operators applying pesticides is a crucial step when approving pesticides for use in Europe. The current method is based on a deterministic exposure model, the AOEM. Exposure calculations are based on a limited number of tests and observations that are conducted and reported only by pesticide companies. Moreover, the number of parameters considered and the scenarios studied are limited, which make the use of the AOEM timely and easy. However, the question of how the parameters are chosen is very important and constitutes a major challenge, since they must accurately cover a range of scenarios and crops. They must ensure the safety of all European workers, regardless of how pesticides are used. As the AOEM is based on studies conducted by the industry under controlled conditions, it is crucial to test whether the predicted values match those observed in usual working conditions. Several academic studies have produced exposure measurements in different countries, for different types of crops and under different working conditions [41]. To our knowledge, however, they have not been compared to date and we are not aware of any independent academic data that could have been integrated in the model. Therefore, we conclude that despite providing an overall overestimation of operator exposure during spraying days in fruit-growing, the AOEM underestimates exposure in some situations and it does not provide an accurate estimate of exposure in some real working conditions that carry a high risk of exposure. This work will be continued on operators in other crop types and on workers who re-enter treated fields to perform tasks including harvesting.

## Figures and Tables

**Figure 1 ijerph-19-04611-f001:**
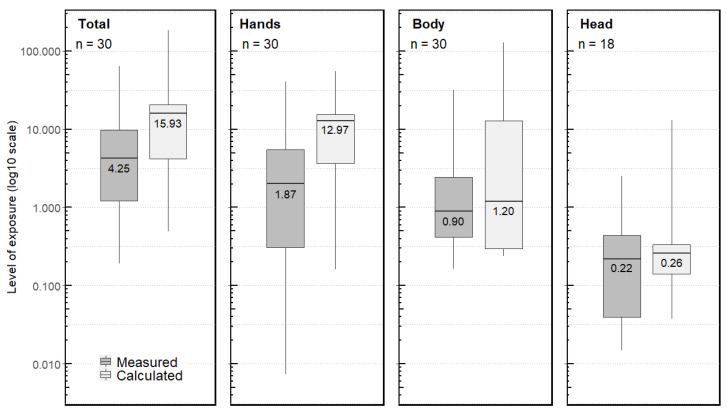
Daily exposure (mg/day) overall (total: hands + body + head) and according to body parts, calculated using the AOEM and measured in the CANEPA study.

**Figure 2 ijerph-19-04611-f002:**
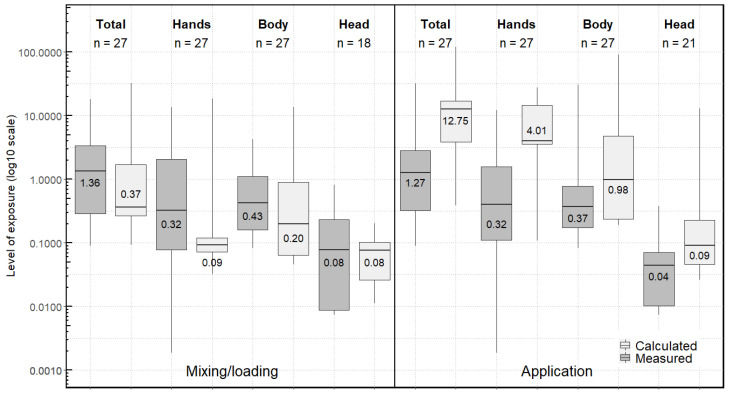
Exposure (mg/day) during mixing/loading and during application, overall (total: hands + body + head) and according to body parts, calculated using the AOEM and measured in the CANEPA study.

**Figure 3 ijerph-19-04611-f003:**
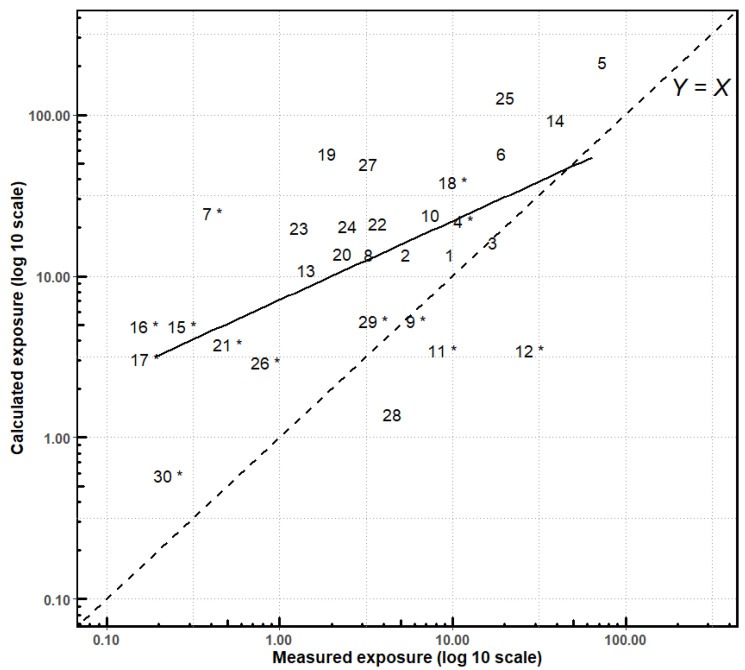
Linear regression of daily exposure (mg): calculated using the AOEM and measured in the CANEPA study. N = 30. Correlation: r (Pearson coefficient) = 0.76, *p*-value < 0.0001. Regression: R^2^ = 0.58. Missing values for head exposure were considered to be zero. Each number corresponds to one observation. * Observations in which dithianon was applied; the others correspond to the application of captan.

**Figure 4 ijerph-19-04611-f004:**
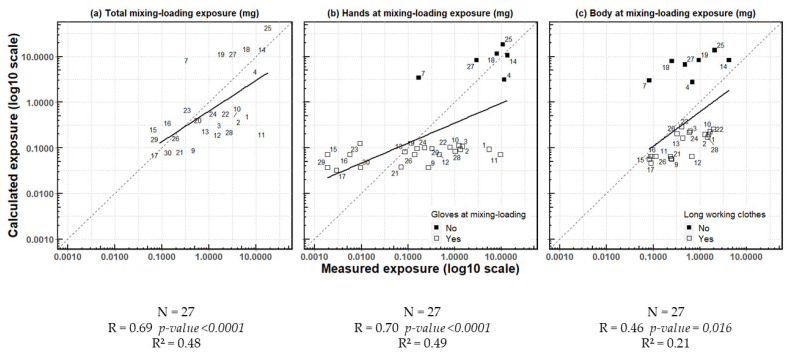
Linear regression of exposure calculated using the AOEM and exposure measured in the CANEPA study during mixing/loading and according to body parts.

**Figure 5 ijerph-19-04611-f005:**
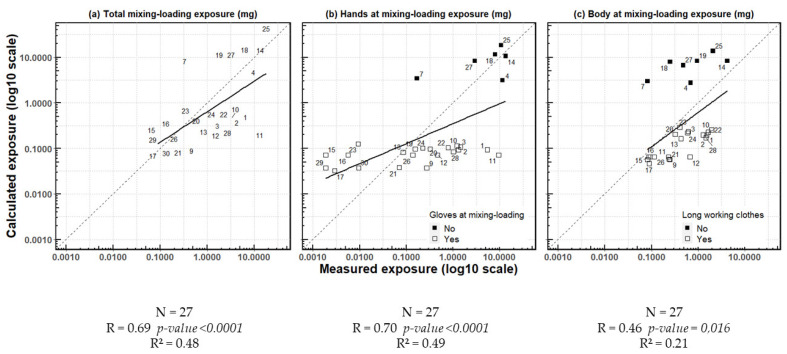
Linear regression of exposure calculated using the AOEM and exposure measured in the CANEPA study during application and according to body parts.

**Table 1 ijerph-19-04611-t001:** Daily amount of active substance applied and daily treated area.

			N	Min	25th Centile	Median	75th Centile	Max
Daily area treated (ha)	In CANEPA		30	1.01	3.26	5	6.9	18.3
	In the AOEM		30	10				
Rate of active substance applied per ha (kg/ha)	Captan WG		17	1.11	1.44	1.52	1.76	3.84
	Dithianon WG		7	0.28	0.33	0.35	0.35	0.36
	Dithianon SC ^1^		6	0.31				
Amount of active substance applied in a day (kg/day)	Captan WG	Applied in CANEPA	17	2.24	5.15	10.32	12	29.6
		Calculated by AOEM		11.1	14.4	15.2	17.6	38.4
	Dithianon WG	Applied in CANEPA	7	0.49	0.61	1.4	2.21	3.36
		Calculated by AOEM		2.8	3.3	3.5	3.5	3.6
	Dithianon SC ^2^	Applied in CANEPA	6	0.63	1.25	1.41	1.56	2.5
		Calculated by AOEM		3.1				

WG = water-dispersible granule. SC = soluble concentrate. ^1^ The rate of dithianon SC applied was the same for the six observations (0.31 kg/ha). ^2^ The amount of dithianon SC applied in a day varied in CANEPA from 0.63 to 2.5 kg/day according to the daily treated area, whereas it did not vary in the estimates of the AOEM because the treated area was fixed at 10 ha.

## Data Availability

Interested persons can contact the corresponding author directly for questions relating the data.

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
