# Peer review of "Pesticide Exposure in Fruit-Growers: Comparing Levels and Determinants Assessed under Usual Conditions of Work (CANEPA Study) with Those Predicted by Registration Process (Agricultural Operator Exposure Model)"

_ijerph, 2022, doi:10.3390/ijerph19084611_

Round 1

Reviewer 1 Report

Dear authors,
I read your manuscript with interest and curiosity. Overall, I think it's a good job. The English language is correct and the working methodology is appropriate. Just a small suggestion in the introductory part of the manuscript, in fact, I think it might be useful to write some more information on the characteristics of pesticides and their danger to professionally exposed workers.
In the guidelines it is advised not to exceed 200 words, your manuscript counts 224. If possible, I suggest trying to find within the limit of 200 words.

Author Response

We would like to thank you for your careful reviews and valuable comments. Changes have been made to the article based on your comments and here is a point by point response.

Just a small suggestion in the introductory part of the manuscript, in fact, I think it might be useful to write some more information on the characteristics of pesticides and their danger to professionally exposed workers.

We improved the introduction with more information on the characteristics of pesticides and their dangers for the professionals exposed to the pesticides at work, in the introduction.

“Pesticide use for crop protection in agriculture developed in the 1930s and in-creased worldwide in the second part of the 20th century (2.3 billion tons in 1990 vs. 4.1 billion tons in 2018) (FAOSTAT). Europe is a major agricultural producer and pesticide consumer with 500,000 tons sold in 2018, including 85,000 only for France (FAOSTAT). There are currently more than 1,000 active substances on the market used in the control of pests, each with different chemical characteristics, targets and modes of action. They can be classified into three main categories: herbicides, insecticides and fungicides. Fungicides were the best-selling pesticide categories in 2018 representing 40% of total pesticides sold in Europe (FAOSTAT) [1]. Understanding the mechanisms of action of these substances also allows us to better understand how they can impact human health. Epidemiological studies have also shown that the use of some of these substances is associated with the occurrence of chronic diseases such as certain cancers, neurodegenerative diseases and respiratory diseases [2–5]. These substances have been classified by the International Agency for Research on Cancer (IARC) into four categories according to the level of evidence of their potential carcinogenicity. Exposure to pesticides poses risks to human health, particularly for agricultural workers in direct contact with them. In 2010 in France, more than one million farmers and farm-workers were exposed to pesticides, to which must be added hundreds of thou-sands of seasonal workers and trainees [6].”

In the guidelines it is advised not to exceed 200 words, your manuscript counts 224. If possible, I suggest trying to find within the limit of 200 words.

We have reduced the abstract to 203 words to fit within the limits set by the publisher.

“Knowledge of pesticide exposure levels in farmers is needed for epidemiological studies and regulatory purpose. In the European pesticide registration process, operators’ exposure is predicted by the Agricultural Operator Exposure Model (AOEM), created in 2014 by the European Food Safety Authority based on studies conducted by the pesticide industry. We compared operators’ exposures during treatment days in apple-growing under non-controlled working conditions and AOEM predicted values. Dermal exposure of thirty French apple-growers from the CANEPA study applying two fungicides was measured by body patches and cotton gloves. For each observation, the corresponding exposure was calculated by the AOEM, using data recorded about the operator, spraying equipment and personal protective equipment (PPE) used. A significant linear correlation was observed between calculated and measured daily exposures. The model overestimated about 4-fold the daily exposure and 10-fold exposure during application. However, exposure was underestimated during mixing/loading for 70% of the observations, when the operator wore PPE. The AOEM did not overestimate exposures in all circumstances, especially during mixing/loading, when operators handle concentrated products. The protection provided by PPE appeared overestimated. This could be due to the optimal working conditions under which the "industrial" studies are conducted, not representative of real working conditions of operators in fruit-growing.”

Reviewer 2 Report

  1. Line 98-99: Consider the sentence, “Operators were...employees.” How does this fact affect the experimentation or its results? The relationship of this information with the degree of precision in the methodology or with the results, if any, may be mentioned.
  2. Line 254-255: Consider the sentence, “In this example...application.” It is not clear whether the operator wore gloves and a mask along with the long working clothes during the process of application also. Please clarify.
  3. Please specify the fabric/material of the long working clothes, if possible, to determine the approximate porosity of the covering.
  4. It would be better if you please explain the logic behind using the median of the data for various parameters to justify the role of the statistical tool; and also state how this explains the observations better than the means of the data.
  5. A few words on the language and grammar of the manuscript: The language and grammar of the manuscript need proper attention. For example, please refer to line 372: The term “data” is plural (of “datum”) and hence be treated accordingly while formulating a sentence. Another example is that of the sentence starting from line 416: “They observed...exposure.” Here “among” or “between” (as the case may be) seems to be omitted between “difference” and “values”. Similarly, typographic errors such as that in line 385 (missing a space between “i.e.” and “approximately); in line 433 (...thus appears thus...) and in line 500 (...had not ‘be’ demonstrated...) must be addressed properly.

Author Response

We would like to thank you for your careful reviews and valuable comments. Changes have been made to the article based on your comments and here is a point by point response.

  • Line 98-99: Consider the sentence, “Operators were...employees.” How does this fact affect the experimentation or its results? The relationship of this information with the degree of precision in the methodology or with the results, if any, may be mentioned.

In response to your comment, we specified in the article that our participants, who are farm owners or permanent employees, had some experience with treatment tasks, had all received agricultural training and obtained the Certiphyto, and were likely aware of the risks associated with pesticides.

Indeed, it is mostly the farm owners and permanent workers who carry out the crop treatment tasks, to which are often added seasonal workers, family helpers, trainees to carry out other tasks (pruning, installation of anti-hail nets, thinning, harvesting) requiring more labor.

Moreover, we did not find any difference in daily exposure levels between farm owners and permanent employees (Wilcoxon test).

  • Line 254-255: Consider the sentence, “In this example...application.” It is not clear whether the operator wore gloves and a mask along with the long working clothes during the process of application also. Please clarify.

We clarified the PPE worn by the operator during mixing/loading and application, in the example given in the article: “In this example, the operator wore gloves, mask and long working clothes during mixing/loading; he only kept long working clothes during application.”

  • Please specify the fabric/material of the long working clothes, if possible, to determine the approximate porosity of the covering.

We added some information on the materials of body PPE in the article: “long cotton working clothes […] and chemical protective suits (Tyvec ®)”

  • It would be better if you please explain the logic behind using the medianof the data for various parameters to justify the role of the statistical tool; and also state how this explains the observations better than the means of the data.

We added in the "statistical analysis" paragraph: “We described the different variables using the median because they were not normally distributed, according to the Shapiro-Wilk test.”

Below is a table showing the means (and standard deviation), medians and Shapiro-Wilk test results for the different variables studied in this article.

Measured variables

Mean ± standard deviation

Median

Shapiro-Wilk test

Area treated (ha)

5.79 ± 4.02

5.00

W = 0.862 (p = 0.0011)

Total amount applied (mg/day)

7.04 ± 7.69

3.68

W = 0.795 (p < 0.0001)

Rate applied (mg/ha)

1.14 ± 0.90

1.20

W = 0.828 (p = 0.0002)

Daily exposure (mg)

9.38 ± 13.85

4.25

W = 0.669 (p < 0.0001)

Hand exposure (mg)

5.45 ± 8.84

1.87

W = 0.645 (p < 0.0001)

Body exposure (mg)

3.68 ± 7.14

0.90

W = 0.537 (p < 0.0001)

Head exposure (mg)

0.39 ± 0.59

0.22

W = 0.621 (p < 0.0001)

Mixing/loading exposure (mg)

3.45 ± 4.78

1.36

W = 0.726 (p < 0.0001)

Application exposure (mg)

3.68 ± 6.86

1.27

W = 0.56 (p < 0.0001)

  • A few words on the language and grammar of the manuscript: The language and grammar of the manuscript need proper attention. For example, please refer to line 372: The term “data” is plural (of “datum”) and hence be treated accordingly while formulating a sentence. Another example is that of the sentence starting from line 416: “They observed...exposure.” Here “among” or “between” (as the case may be) seems to be omitted between “difference” and “values”. Similarly, typographic errors such as that in line 385 (missing a space between “i.e.” and “approximately); in line 433 (...thus appears thus...) and in line 500 (...had not ‘be’ demonstrated...) must be addressed properly.

We corrected the grammatical and linguistic inaccuracies you pointed out.

Reviewer 3 Report

This is excellent paper addressing an important aspect of pesticide regulation.  The authors provide a solid justification of their study, and provide a strong description of their data collection and analytic methods.  The presentation of results is straightforward.  The discussion, particularly of external validity, is informative.

My suggestions are limited.

Lines 126-131:  The measures are listed, but not defined.

Line 553:  Some explanation of why IRB review was not applicable should be provided.

Minor

Line 84: CANEPA – should be defined when first used  

Line 528: change “timeliness” to “timely”

Author Response

We would like to thank you for your careful reviews and valuable comments. Changes have been made to the article based on your comments and here is a point by point response.

  • Lines 126-131:  The measures are listed, but not defined.

All these variables were described for our study population in the article: Bureau et al. "Pesticide exposure of workers in apple growing in France." published in Int Arch Occup Environ Health, in 2021. https://doi.org/10.1007/s00420-021-01810-y.

We have added this information to the article and cited the article by Bureau et al. (2021).

  • Line 553:  Some explanation of why IRB review was not applicable should be provided.

When the study was conducted in 2016-2017, consent to participate was sufficient for an observational study of routine work tasks, according to French law. In France, ethics committees are required for all research involving the human person in the view of developing biological or medical knowledge (L 1123-7 du Code de la Santé Publique). In the present study, no experimentation was performed, only observation of the usual conditions of work, no medical data was collected, no health care was provided and no drug use. Thus, there was no formal requirement for obtaining authorization from an ethics committee. However, we chose to provide detailed and standardized information to the participants: they were given written information on the study, and we asked them to sign a consent form to participate.

  • Line 84: CANEPA – should be defined when first used  

CANEPA means CANcer and Exposure to Pesticides in Agriculture. We added it in the article.

  • Line 528: change “timeliness” to “timely”

We corrected this inaccuracy.